# Oil Palm AP2 Subfamily Gene *EgAP2.25* Improves Salt Stress Tolerance in Transgenic Tobacco Plants

**DOI:** 10.3390/ijms25115621

**Published:** 2024-05-22

**Authors:** Lixia Zhou, Hongxing Cao, Xianhai Zeng, Qiufei Wu, Qihong Li, Jerome Jeyakumar John Martin, Dengqiang Fu, Xiaoyu Liu, Xinyu Li, Rui Li, Jianqiu Ye

**Affiliations:** 1National Key Laboratory for Tropical Crop Breeding, Chinese Academy of Tropical Agricultural Sciences, Haikou 571101, China; lxzhou@catas.cn (L.Z.); caohx@catas.cn (H.C.); zxh200888@126.com (X.Z.); qfi_wu@catas.cn (Q.W.); liqihong@catas.cn (Q.L.); jeromejeyakumarj@gmail.com (J.J.J.M.); fudq@catas.cn (D.F.); liuxy86@catas.cn (X.L.); lixinyu@catas.cn (X.L.); 2Coconut Research Institute, Chinese Academy of Tropical Agricultural Sciences, Wenchang 571339, China

**Keywords:** *EgAP2.25* gene, oil palm, tobacco, salinity stress, physiological and biochemical indexes, stress marker genes

## Abstract

AP2/ERF transcription factor genes play an important role in regulating the responses of plants to various abiotic stresses, such as cold, drought, high salinity, and high temperature. However, less is known about the function of oil palm AP2/ERF genes. We previously obtained 172 AP2/ERF genes of oil palm and found that the expression of *EgAP2.25* was significantly up-regulated under salinity, cold, or drought stress conditions. In the present study, the sequence characterization and expression analysis for *EgAP2.25* were conducted, showing that it was transiently over-expressed in *Nicotiana tabacum* L. The results indicated that transgenic tobacco plants over-expressing *EgAP2.25* could have a stronger tolerance to salinity stress than wild-type tobacco plants. Compared with wild-type plants, the over-expression lines showed a significantly higher germination rate, better plant growth, and less chlorophyll damage. In addition, the improved salinity tolerance of *EgAP2.25* transgenic plants was mainly attributed to higher antioxidant enzyme activities, increased proline and soluble sugar content, reduced H_2_O_2_ production, and lower MDA accumulation. Furthermore, several stress-related marker genes, including *NtSOD*, *NtPOD*, *NtCAT*, *NtERD10B*, *NtDREB2B*, *NtERD10C*, and *NtP5CS*, were significantly up-regulated in *EgAP2.25* transgenic tobacco plants subjected to salinity stress. Overall, over-expression of the *EgAP2.25* gene significantly enhanced salinity stress tolerance in transgenic tobacco plants. This study lays a foundation for further exploration of the regulatory mechanism of the *EgAP2.25* gene in conferring salinity tolerance in oil palm.

## 1. Introduction

Salinization is one of the biggest environmental considerations that seriously affects seed germination, crop growth, and yield [1,2]. Salinization has adverse effects on agricultural land, leading to the decline of soil fertility, and has posed a serious threat to global crop production [3]. In the world, the total area of saline soil is 954.38 million hectares, which is increasing every year [4]. Plants have evolved different stress response and tolerance regulation pathways, such as hormone signaling [5], salt stress signal transduction [6], chromatin modification [7], and transcriptional regulation [8]. Transcription factors are a class of protein molecules with special structures that can regulate gene expression. They specifically recognize and bind to the cis-acting elements in the upstream promoter region of related genes or other proteins, ensuring that the target gene is expressed in a specific time and space [9]. Plants require a larger number of specific transcription factors to regulate functional genes in response to salt stress, which can be divided into different gene families, such as WRKY, NAC, MYB, bZIP, AP2/ERF, bHLH, and other transcription factor families. As regulation and molecular switches, they play a crucial role in the regulation network of salt stress signals [10,11,12,13,14,15,16]. Additionally, under salt stress, plants produce a large amount of reactive oxygen species (ROS), which have a negative impact on plant growth and development [17,18].

The AP2/ERF superfamily is one of the largest gene families in plants, and it has at least one AP2 domain composed of around 60 amino acids [19]. According to the types and quantities of structural domains, they can be divided into AP2, RAV, ERF, DREB, and Soloist [20]. Interestingly, AP2/ERF genes participate in plant growth, development and apoptosis, hormone signaling pathway transduction, and biotic and abiotic stress responses [21]. *TgERF1*, from *Tectona grandis*, improved tolerance to salt stress in tobacco. The improved stress tolerance of transgenic plants compared to control plants was supported by morphological and physio-biochemical data, and by activation of an array of stress-responsive genes analyzed at the molecular level, leading to the synthesis of protective compounds [22]. *PvERF35* was amplified from the common bean (*Phaseolus vulgaris* L.), cloned, and functionally characterized by overexpressing in tobacco. The physiological and biochemical analysis of transgenic plants showed their better performance compared to the wild-type in terms of germination, survival rates, and root and shoot growth under salt stress treatment (200 mM NaCl). Having a high concentration of proline, APX, and POX, the *PvERF35* overexpressed plants were physiologically and morphologically less affected by salt stress application [23]. In addition, *GmDREB1*, as an AP2/ERF transcription factor, enhanced the drought resistance of transgenic soybeans by interacting with *GmERFs* [24]. Overexpression of *OsEREBP1* and *SmERF1* improved salt tolerance in transgenic rice and transgenic eggplant, respectively [25]. Furthermore, *ZmEREB57* enhanced the salt stress tolerance of maize by binding to the promoter of *ZmAOC2* [26]. However, there is still limited knowledge regarding the functional validation of oil palm AP2/ERFs, and further research is necessary to explore the salinity stress signaling pathway in oil palm.

Oil palm (*Elaeis guineensis*) is mainly used for palm oil production and is one of the main woody economic crops in tropical regions [27]. Nevertheless, palm oil productivity is significantly affected by environmental factors, such as extreme weather, salinity, and drought [28,29,30]. Previously, we finished the identification and characterization of oil palm AP2/ERF family genes and analyzed the expression of some oil palm AP2/ERF genes under abiotic stress conditions with real-time quantitative PCR (qPCR) [14]. We also obtained *the EgAP2.25* gene, which was induced by salinity stress. The 5′ region of the oil palm *uep1* gene, which contained an 828 bp sequence upstream of the *uep1* translational start site, were isolated and characterized. Construction of a pUEP1 transformation vector, which contained the *gusA* reporter gene under the control of the *uep1* promoter, was carried out for functional analysis of the promoter through transient expression studies. It was found that the 5′ region of *uep1* functioned as a constitutive promoter in oil palm and could drive GUS expression in tobacco, which was used as a model plant for transformation [31,32]. In this study, the *EgAP2.25* was cloned from the African oil palm (*E. guineensis* Jacq.), and its sequence characterization and expression patterns were analyzed. In addition, we analyzed the antioxidant enzyme activities, osmolytes (MDA, proline, and soluble sugar), chlorophyll, and electrolyte leakage in transgenic tobacco plants in salinity conditions to study the function of *EgAP2.25* for protecting the plants from salinity stress. Finally, a functional mechanism model of the *EgAP2.25* gene in response to salinity stress was depicted. The research will provide possibilities for clarifying the salt stress tolerance mechanism of oil palm and exploring the role of AP2/ERF genes in other crops.

## 2. Results

### 2.1. Sequence and Character of Oil Palm EgAP2.25

We previously finished the identification of 172 oil palm AP2/ERF genes at the genomic level [14]. Here in our study, the sequence feature and expression pattern of oil palm *EgAP2.25* were analyzed. There were ten conserved motifs and two AP2/ERF conservative domains in the *EgAP2.25* protein, which were located at 197–266 aa and 301–360 aa, respectively (Figure 1a,c). The *EgAP2.25* gene contained 7 exons and 6 introns (Figure 1b). Furthermore, by predicting the 2000 bp promoter sequence upstream of the *EgAP2.25* gene, we obtained *cis*-acting elements related to stress (P-box, TATC-box, CGTCA-motif, GC-motif, and TGACG-motif) and plant growth (G-Box, G-box, AT1-motif, ATCT-motif, GATA-motif, and TCCC-motif) (Figure 1d). By predicting and analyzing the tertiary structure of *EgAP2.25* protein, it was found that the protein had 7 transmembrane domains (TMs) with hormone-related motifs between TM2 and TM3 (Figure 1e). Moreover, *EgAP2.25* significantly over-expressed at 24 h under salt conditions (Figure 1f).

### 2.2. Generation of Transgenic Tobacco Plants Over-Expressing EgAP2.25

The transgenic tobacco plants over-expressing *EgAP2.25* were obtained (Figure 2). The independent transgenic lines were selected based on high transgene expression levels. T1~T3 generations were selected on a kanamycin selection medium and screened via PCR analysis using a gene-specific primer of *EgAP2.25* (Figure 3a). Expression analysis of *EgAP2.25* was further confirmed in independent transgenic lines through reverse transcriptional PCR (RT-PCR) analysis. In addition, four *EgAP2.25* transgenic tobacco plants (T0, T1, T2, and T3) over-expressing *EgAP2.25* were checked by semi-quantitative RT-PCR (Figure 3b).

### 2.3. Analysis of Salinity Stress on the Germination Index and Phytomass of Tobacco Lines

Seed germination is the beginning of the whole life process, and the germination period is sensitive to salinity conditions. Germination rate, root length, and dry and fresh weight can be characterized as effective indicators for salinity tolerance screening during germination. Wild-type (WT) and *EgAP2.25* transgenic tobacco seeds were grown under 0 mM, 100 mM, 200 mM, and 300 mM NaCl, respectively. The results showed that all WT and transgenic plants grew well under normal watering conditions. At 100 mM NaCl, *EgAP2.25* transgenic plants exhibited a germination rate of 97% compared to 84% in WT lines. Under 200 mM NaCl stress, the germination rate of transgenic plants (76%) was significantly higher than that of WT plants (43%). Notably, at 300 mM NaCl, the germination rate of WT lines was markedly lower (19%) than that of transgenic plants (70%) (Figure 4a,b). Moreover, with the salt concentration increased, the root lengths of WT and transgenic lines decreased significantly. However, the root lengths of T1, T2, and T3 lines were measurably longer (*p* < 0.05) than that of WT lines at 200 mM NaCl. When the salt concentration reached 300 mM, the root lengths decreased significantly, but the transgenic lines were longer than that of WT (Figure 4c and Figure 5a).

In addition, the changes in dry/fresh weight of WT and transgenic lines under different salinity were investigated. There was no significant difference in dry/fresh weight between WT and transgenic plants under normal watering conditions. However, when the salt concentration reached 200 mM, the dry/fresh weight of transgenic lines was significantly higher than that of WT lines (Figure 4d,e). Subsequently, as the salt concentration rose to 300 mM, both transgenic and WT lines experienced a significant decrease in weight, with the dry/fresh weight of transgenic tobacco plants becoming comparable. Further, a salinity tolerance analysis was conducted on potted plants, revealing that after 5 days of recovery in normal watering conditions, the *EgAP2.25* transgenic tobacco plants showed healthy green leaves, whereas the WT plants wilted and approached death (Figure 5b). These results proved that over-expression of *EgAP2.25* enhanced the salinity tolerance of transgenic plants.

### 2.4. Antioxidant Enzyme Activities in EgAP2.25 Transgenic Tobacco

In order to evaluate whether over-expression of *EgAP2.25* could improve the antioxidant enzyme activities, which can protect transgenic tobacco plants from salinity stress, WT and *EgAP2.25* transgenic tobacco plants were subjected to varying concentrations of NaCl (0 mM, 100 mM, 200 mM, and 300 mM). There were no differences under the 0 mM NaCl condition (Figure 6). However, under salinity stress, the superoxide dismutase (SOD), peroxidase (POD), and catalase (CAT) activities of *EgAP2.25* transgenic plants were significantly increased compared to WT plants. Specifically, at 100 mM NaCl, the SOD activity of transgenic plants was higher (23.08~34.20%, *p* < 0.05) than that of WT plants, and the POD and CAT activities were both significantly elevated (62.29~64.20% and 54.36~57.10%, *p* < 0.01). With a further increase in NaCl concentration to 200 mM, both SOD and POD activities were significantly higher (40.68~41.54%, 40.31~42.33%, *p* < 0.01) compared to WT plants.

### 2.5. Physiological and Biochemical Variation in EgAP2.25 Transgenic Tobacco

Malondialdehyde (MDA) is a peroxide product of cell membranes, and changes in its content serve as an indicator to assess alterations in cell membrane permeability in plants under salinity conditions. Content of MDA in transgenic tobacco was notably lower (9.325~10.312 µmol·g^−1^ FW) compared to WT (16.810 µmol·g^−1^ FW) under 100 mM NaCl condition, and 16.953–18.886 µmol·g^−1^ FW of MDA formation compared to WT (24.673 µmol·g^−1^ FW) at 200 mM NaCl (Figure 7a). Additionally, the H_2_O_2_ content was conducted to evaluate the oxidative damage in WT and over-expression lines. The results revealed that under 200 mM NaCl salinity stress, the H_2_O_2_ content increased in all WT and transgenic plants compared to the control (0 mM NaCl), but all over-expression lines (4.841~5.101 µmol·g^−1^ FW) showed significantly lower H_2_O_2_ content compared to WT lines (7.401 µmol·g^−1^ FW) (Figure 7b).

Proline, as a crucial osmoregulatory substance in plant cells, was investigated in WT and *EgAP2.25* transgenic tobacco plants under salinity conditions. Initially, there was no difference between WT and transgenic plants under 0 mM NaCl. However, upon exposure to 200 mM NaCl, proline content in transgenic tobacco plants was remarkably increased by 4~6 fold compared to that in WT (Figure 7c). Moreover, the levels of soluble sugars followed a similar trend to proline content. Specifically, no disparity was observed between the over-expression and WT plants under normal watering condition. Yet, following treatment with 200 mM NaCl, the content of soluble sugar in transgenic lines increased by 62.604% to 64.491% (Figure 7d).

Chlorophyll degradation in plants not only directly relates to the photosynthetic process, but also serves as a crucial physiological marker for determining plant salinity tolerance. To explore this further, the chlorophyll content of WT and *EgAP2.25* transgenic tobacco plants was examined under varying NaCl concentrations. The results indicated that chlorophyll content of all tobacco lines increased at various NaCl concentrations compared to the control (0 mM NaCl). The over-expression lines showed higher chlorophyll (21.232~21.641%) and (36.584~38.137%) compared to WT plants, under 200 and 300 mM NaCl concentrations (Figure 7e), respectively. Additionally, the measurement of relative electrolytic leakage from plant leaves acts as a crucial signal of the extent of damage to plant cell membranes. The results demonstrated that the over-expression lines displayed lower electrolytic leakage (13.832~14.503%) and (26.447~27.196%) compared to WT plants, at under 200 and 300 mM NaCl concentrations (Figure 7f).

### 2.6. Expression Profiles of EgAP2.25 and Salinity Stress Marker Genes in Transgenic Tobacco Plants

*EgAP2.25* expression profiles in different tissues (root, stem, leaf, and flower of *EgAP2.25* transgenic tobacco) were explored by qPCR. The results showed significantly different expressions in different organizations of transgenic tobacco lines (Figure 8). Furthermore, the expression of salinity stress marker genes such as *NtSOD*, *NtPOD*, *NtCAT*, *NtERD10B*, *NtDREB2B*, *NtERD10C*, and *NtP5CS* was analyzed in over-expressed and WT tobacco plants under 200 mM NaCl. Results reflected the significant over-expression of all tested salinity stress marker genes in transgenic tobacco under salinity conditions exposure compared to WT lines (Figure 9). Thus, the results indicated that *EgAP2.25* enhanced the up-regulation of stress-related genes to improve the salinity tolerance in transgenic tobacco plants.

## 3. Discussion

AP2/ERF is one of a superfamily of transcription factors widely existing in plants, which can participate in regulating plant responses to environmental stresses and stimulate the interest of researchers [19]. Previously, we obtained 172 AP2/ERF factors from the oil palm genome [14] and confirmed that the expression of some oil palm AP2/ERF genes increased significantly under salinity stress treatment. On the basis of the previous research results, the present study focused on *EgAP2.25*, a member of the AP2 subfamily, by over-expressing it in *Nicotiana tabacum* L. All results indicated that the over-expression of the *EgAP2.25* gene could improve the salinity tolerance of transgenic tobacco plants (Figure 10).

Previous studies have found that the over-expression of AP2/ERF genes such as *AtERF71/HRE2*, *PvERF35*, and *MaDREB20* can increase the abiotic stress tolerance in transgenic plants [23,33,34]. In accordance with this, our results also confirmed that over-expression of the *EgAP2.25* gene can improve the salinity tolerance of transgenic tobacco plants. Under salinity conditions, reactive oxygen species (ROS) content in plant cells increases rapidly. In order to slow down the damage of ROS to the metabolism balance and cell plasma membrane, plants will start the antioxidant defense system to remove excess ROS [35]. As the main enzymes in the plant antioxidant system, SOD, POD, and CAT could protect the plants from harsh abiotic stress [36,37]. In this study, the enzyme activities of SOD, POD, and CAT were stronger in *EgAP2.25*-over-expressed tobacco plants compared to that of WT lines. This result was consistent with previous results describing that over-expression of *PagERF072* enhanced the SOD, POD, and CAT enzyme activities in over-expressed poplar plants subjected to salinity stress [38]. According to the results, we could speculate that the improvement of salinity tolerance of *EgAP2.25* transgenic tobacco is related to the enhancement of antioxidant enzyme activities.

Under normal conditions, ROS produced in plants are at a low level and do not harm the plants. However, during salt stress, the metabolic balance of plants is disrupted. The increase in ROS causes the degradation of protein and chlorophyll [39]. In our study, the content of chlorophyll in *EgAP2.25*-over-expressed tobacco plants was higher as compared to WT lines under salinity conditions, suggesting that the photosynthetic apparatus of over-expression lines suffered less damage and had stronger photosynthetic capacity. Proline and soluble sugar are important osmoregulation substances in plants, which can increase the concentration of cytoplasm, maintain the osmotic potential of cells, increase the hydration of protein molecules, maintain photosynthetic characteristics, and reduce the damage of oxidative stress to cells [40]. Transgenic tobacco accumulated more proline and soluble sugar than WT lines under salinity stress conditions, which was more conducive to preventing oxidative stress. Accordingly, our study showed that *EgAP2.25* transgenic tobacco lines have stronger salinity tolerance than WT lines, and under salinity stress conditions, the growth of over-expression plants was better, partly because of the higher content of osmotic fluid compared to WT plants. Moreover, *EgAP2.25* transgenic tobacco plants contained lower content of H_2_O_2_ and MDA than WT lines. This result told us that over-expression tobacco plants suffered less oxidative damage than WT plants. In this study, our results were in agreement with various reports describing that over-expression of *EgAP2.25* enhanced the resistance to salinity stress of transgenic tobacco plants by improving antioxidant enzyme activities, proline, and soluble sugar content and reducing H_2_O_2_ and MDA content [41,42,43].

Seed germination is the initial stage of crop life and is also the most sensitive stage to environmental conditions [44,45]. Studies have shown that the germination rate of seeds is one of the main indicators to measure the tolerant ability of plants to abiotic stresses [46,47]. High percentage of germination is the foundation of agricultural production [48]. Li et al. uncovered *OsSAE1*, as a positive regulator, improved seed germination and salt tolerance of rice [49]. *LcAP2/ERF107*, isolated from *Lotus corniculatus* cultivar Leo, significantly improved the seed germination rate and enhanced salt stress tolerance of transgenic *Arabidopsis* [50], which was consistent with our finding that transgenic tobacco lines over-expressing *EgAP2.25* showed a higher seed germination rate, longer root length, and higher fresh weight and dry weight than WT lines under various NaCl concentrations.

As reported in many studies, the over-expression of AP2/ERF genes could up-regulate the stress-related marker genes in various transgenic plants under various stress treatments [44,45]. In our study, the over-expression of 35S-*EgAP2.25* via qPCR led to the high expression profiles of stress-responsive genes, including *NtSOD*, *NtPOD*, *NtCAT*, *NtERD10B*, *NtDREB2B*, *NtERD10C*, and *NtP5CS* in transgenic tobacco plants under salinity conditions. The results provide possibilities for constructing a regulatory network of the *EgAP2.25* gene in response to salinity stress in oil palm and provide a scientific basis for further elucidating the gene regulatory network of oil palm salinity tolerance.

## 4. Materials and Methods

### 4.1. Plant Materials, Growth Condition, and Salinity Treatment

The African oil palm (*E. guineensis* Jacq.) seedlings were grown in greenhouses (27 °C/temperature; 16 h/light; 8 h/darkness; humidity of about 50~60%) at the Coconut Research Institute, CATAS, Wenchang, China. Healthy 6-month-old oil palm seedlings were selected for salinity treatments. Before the stress of such exposure, all plants were shifted to a growth chamber at 27 °C for one day. Salinity stress was induced by immersing the roots of the oil palm seedlings in 200 mM NaCl. After 48 h, the spear leaves were collected and immediately frozen in liquid nitrogen for further RNA isolation.

*Nicotiana tabacum* L. was used for the experiments. First, the seeds were treated with 75% ethanol for 30 s, then sterilized in 1% NaClO for 15 min. After washing three times with sterile water, the seeds germinated on Murashige and Skoog (MS) medium, and the growth status was observed after sowing.

### 4.2. Cloning and Amplification of EgAP2.25 Gene

Total RNA was isolated from the young leaves of African oil palm plants exposed to salinity stress (300 mM NaCl, 48 h) using Total RNA Extractor (Trizol) [13]. The synthesis of cDNA was achieved using the MightyScript Plus first-strand cDNA synthesis kit. According to the known *EgAP2.25* gene sequence of oil palm, a pair of primers (Forward: 5′-TTGGCTGCATCTGGGAAGAG-3′, reverse: 5′-TCCCGCTCCATTTCAGCTTAG-3′) were used to amplify the open reading frames of *EgAP2.25*. The PCR amplified fragment was cloned into the binary vector pCAMBIA1301 under the control of the CaMV35S promoter and generated pCAMBIA1301-*EgAP2.25* vector. The recombinant plasmid was further mobilized into *Agrobacterium tumefaciens* strain GV3101.

### 4.3. Tobacco Transformation

Transformation of *N. tabacum* was produced as the reported method [51]. Disc leaf explants of 6 mm size from in vitro-grown shoot cultures of *N. tabacum* were used for transformation using *Agrobacterium tumefaciens* strain GV3101 harboring pCAMBIA1301-*EgAP2.25*. After co-cultivation, explants were further transferred to a kanamycin (100 mg·L^−1^) culture medium. The putative transformants were further transferred to the growth chamber and verified with PCR. The putative T0 transformants were further self-pollinated and T1, T2, and T3 seeds were harvested, which were used for subsequent functional analysis.

### 4.4. Salinity Stress Treatment to Transgenic Tobacco Plants

*EgAP2.25* transgenic seeds and WT seeds were sterilized and germinated on the 1/2 MS medium containing 0 mM, 100 mM, 200 mM, and 300 mM NaCl. Green house-grown transgenic and WT seedlings were transferred to well-mixed soil (soil:peat:perlite = 1:1:1) and subjected to salinity stress at the 4-week-old stage by irrigating with water containing 0 mM, 100 mM, 200 mM, and 300 mM NaCl solutions.

In order to measure the ratio of germination, the sprouted seeds were counted every day after sowing. To analyze the seedling growth, root lengths, and fresh/dry weight, 1-week-old seedlings of WT and transgenic plants were shifted to 1/2 MS medium containing 0 mM, 100 mM, 200 mM, and 300 mM NaCl and allowed to grow vertically for the next 4 weeks. On the basis of better seed germination and seedling survival response, the antioxidant enzyme activities and physiological parameters were explored by watering with 0 mM, 100 mM, 200 mM, and 300 mM NaCl, respectively. Further, to explore the phenotype and the expression pattern of the *EgAP2.25* gene in different tissues (root, shoot, stem, and leaf) of WT and transgenic tobacco, all potted seedlings were watered with 200 mM NaCl solution.

### 4.5. Physiological and Biochemical Analyses of WT and EgAP2.25 Transgenic Tobacco

For physiological and biochemical parameter determination, leaves were taken from WT and transgenic plants before and after salinity stress at different salt concentrations. SOD, POD, and CAT activities of WT and *EgAP2.25* transgenic plants were measured with the method described by Liang et al. (2022) [52]. Proline, MDA, soluble sugar, H_2_O_2_, chlorophyll content, and electrolytes involved in salinity stress were measured by methods described previously [17,53].

### 4.6. Expression Analysis of EgAP2.25 and Salinity Stress Marker Genes via Quantitative Real-Time PCR

To explore the expression of *EgAP2.25* in different tissues of transgenic plants, total RNA was extracted from the roots, stems, flowers, and leaves of WT and transgenic plants. To analyze the expression of salinity stress-related genes (*NtSOD*, *NtPOD*, *NtCAT*, *NtERD10B*, *NtDREB2B*, *NtERD10C*, and *NtP5CS*), WT and transgenic tobacco were watered with 200 mM NaCl solution at 0, 1, 2, 3, and 4 days, respectively. RNA extraction and cDNA synthesis were referred according to in the previous report [11]. qRT-PCR was carried out using SYBR Green qPCR Promix (low ROX) protocol in 96-well optical plates with 10 μL reaction volume. The primers of *EgAP2.25*, *NtSOD*, *NtPOD*, *NtCAT*, *NtERD10B*, *NtDREB2B*, *NtERD10C*, *NtP5CS*, and Actin were shown in Appendix A. The oil palm *Actin* gene was used as a housekeeping gene to check the relative expression of *EgAP2.25*, *NtSOD*, *NtPOD*, *NtCAT*, *NtERD10B*, *NtDREB2B*, *NtERD10C*, and *NtP5CS* by 2^−ΔΔCt^ method [54].

## 5. Conclusions

In conclusion, the function of the *EgAP2.25* gene in oil palm under salt stress response was elaborated. It was found that EgAP2.25 contained two conserved AP2/ERF domains and 7 TMs. In addition, the expression level of *EgAP2.25* was highest under salinity stress for 24 h through qRT-qPCR analysis. Over-expression of the *EgAP2.25* gene increased the salinity tolerance of transgenic tobacco plants by improving the activities of SOD, POD, and CAT, increasing the content of proline, soluble sugar, and chlorophyll, reducing the accumulation of MDA, H_2_O_2_, and electrolyte, and up-regulating the expression of several stress-responsive marker genes (*NtSOD*, *NtPOD*, *NtCAT*, *NtERD10B*, *NtDREB2B*, *NtERD10C*, and *NtP5CS*) under salinity treatment. The schematic representation of the background information of the *EgAP2.25* gene was thus depicted. The functional research of the *EgAP2.25* gene will provide novel strategies for salinity-tolerant molecular-assisted breeding of oil palm.

## Figures and Tables

**Figure 1 ijms-25-05621-f001:**
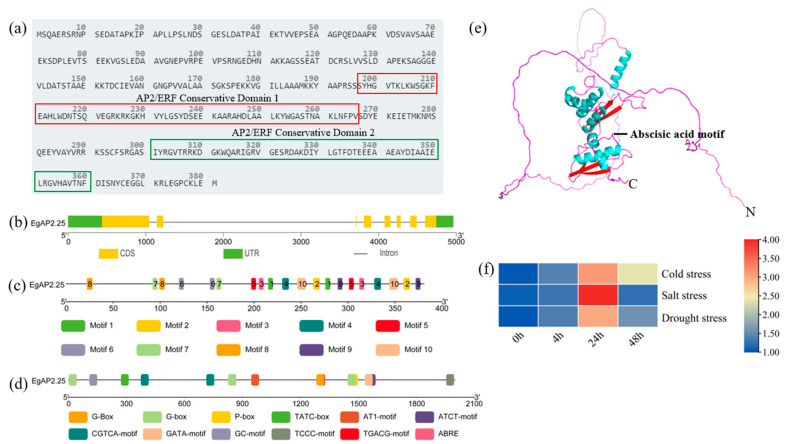
Character of oil palm *EgAP2.25*. (**a**) Conserved AP2/ERF domains of *EgAP2.25* protein. Red box represented conserved domain 1 and green box represented conserved domain 2. (**b**) Structure of *EgAP2.25* gene. (**c**) Motif of *EgAP2.25* protein. (**d**) Distribution and function of *cis*-acting regulatory elements in the promoter of *EgAP2.25*. (**e**) The tertiary structure of *EgAP2.25* protein. Rose red represented random coil, red represented β-sheet and blue represented α-helix. (**f**) Expression of *EgAP2.25* gene under cold, salt, and drought stresses.

**Figure 2 ijms-25-05621-f002:**
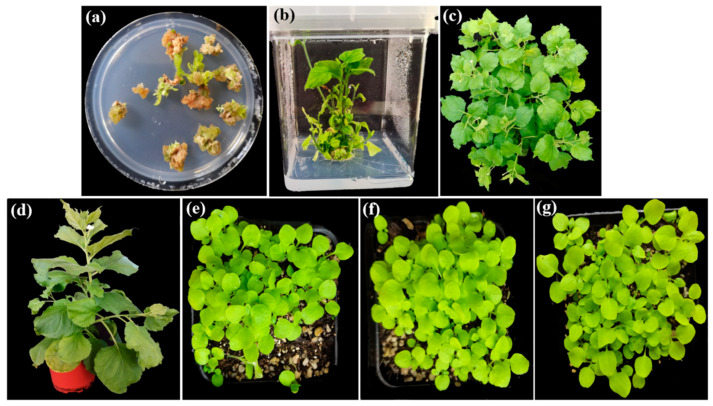
Steps involved in genetic transformation of tobacco. (**a**) Callus formation on media. (**b**) Rooted transgenic plant. (**c**,**d**) Pot-grown transgenic plant (T0 lines). (**e**) T1 transgenic plants. (**f**) T2 transgenic plants. (**g**) T3 transgenic plants.

**Figure 3 ijms-25-05621-f003:**
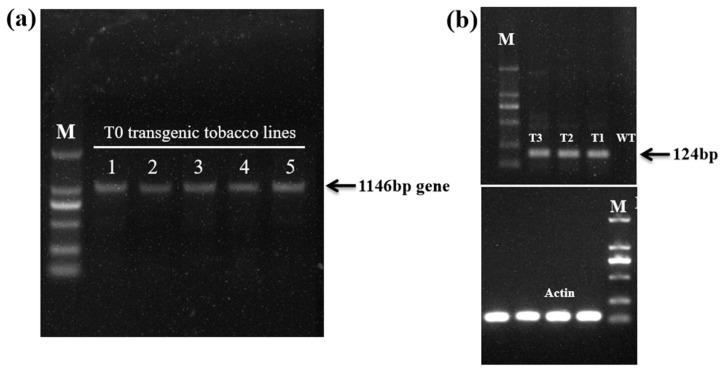
Profile of transgenic tobacco plants. (**a**) PCR results of T0 *EgAP2.25* transgenic tobacco lines. (**b**) Semi-quantitative RT-PCR analysis of *EgAP2.25* (124 bp) gene in WT, T1, T2, and T3 transgenic tobacco lines.

**Figure 4 ijms-25-05621-f004:**
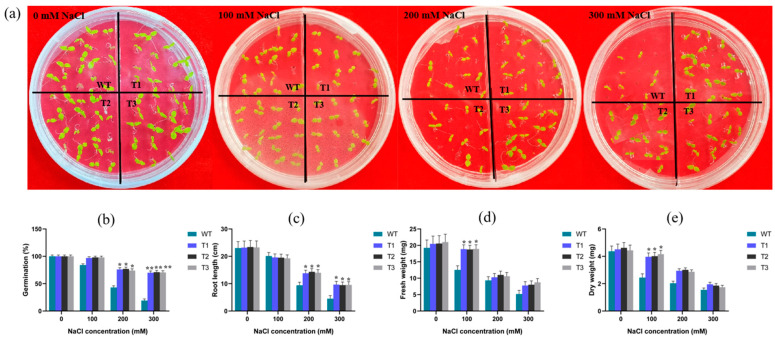
Seed germination and seedling growth of wide and transgenic tobacco. (**a**) Germination of seeds treated with 0, 100, 200, 300 mM NaCl. (**b**–**e**) Germination percentage, root length, and dry/fresh weight of WT and transgenic tobacco at 0~300 mM NaCl, respectively. Numerical values were means ± SD (*n* = 3), ** represents *p* < 0.01, and * represents *p* < 0.05.

**Figure 5 ijms-25-05621-f005:**
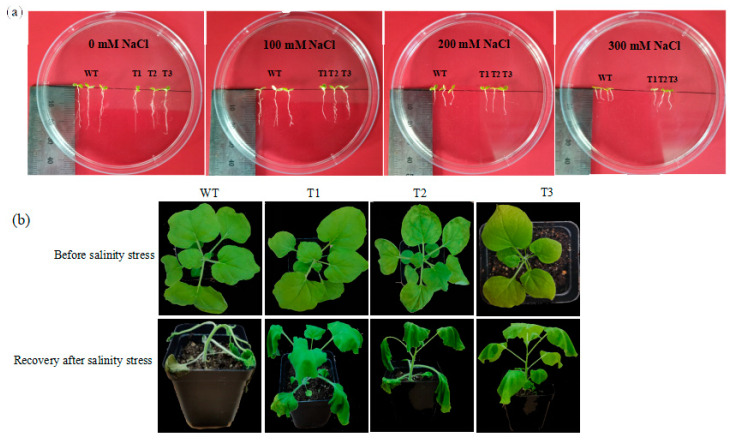
Effect of *EgAP2.25* gene expression under salt stress. (**a**) Comparative observations of root, shoot, and leaf growth between WT and transgenic tobacco lines. (**b**) Phenotypes of WT and transgenic *EgAP2.25* tobacco lines watered with 200 mM of NaCl solution.

**Figure 6 ijms-25-05621-f006:**
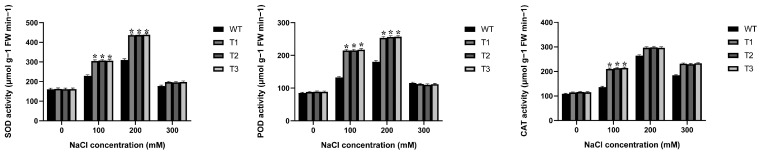
Antioxidant enzyme activity in *EgAP2.25* transgenic and WT tobacco plant leaves under 0, 100, 200, and 300 mM NaCl stress.* represents *p* < 0.05.

**Figure 7 ijms-25-05621-f007:**
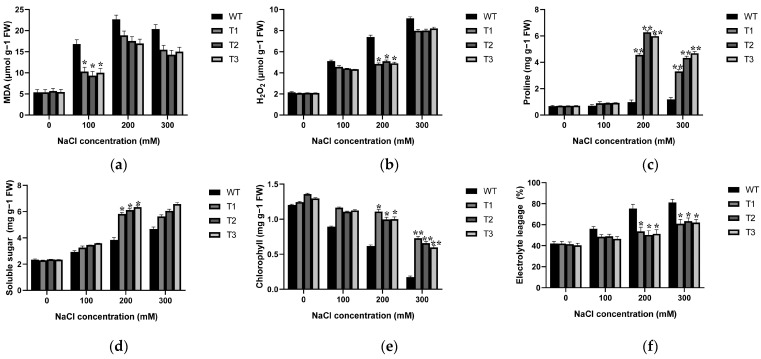
Changes in MDA (**a**), H_2_O_2_ (**b**), proline (**c**), soluble sugar (**d**), chlorophyll content (**e**), and relative conductivity (**f**) in *EgAP2.25*-over-expressed and WT tobacco plants under salt conditions. ** represents *p* < 0.01, and * represents *p* < 0.05.

**Figure 8 ijms-25-05621-f008:**
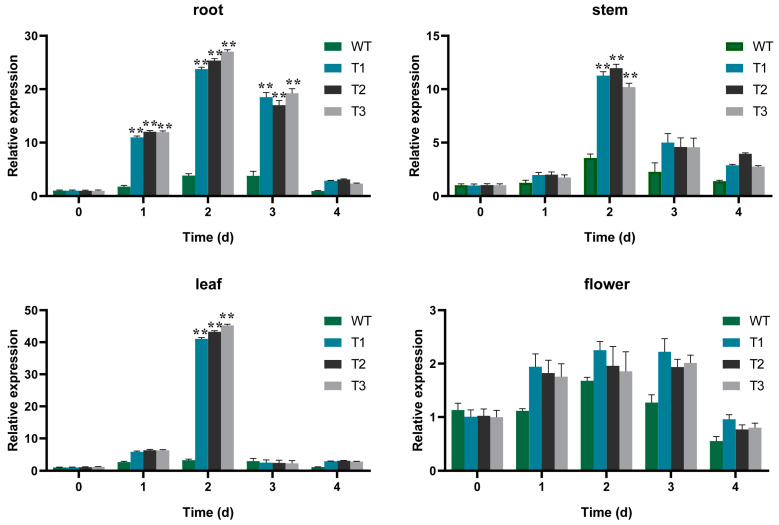
qPCR validation of *EgAP2.25* gene expression in transgenic tobacco tissues with 200 mM NaCl solution. ** means *p* < 0.01.

**Figure 9 ijms-25-05621-f009:**
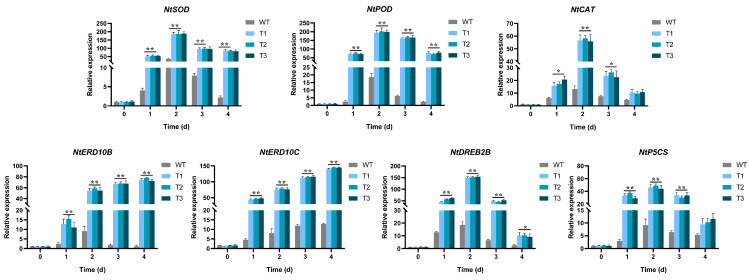
Antioxidant enzyme genes expression in WT and transgenic tobacco with 200 mM NaCl solution. * represents *p* < 0.05 and ** represents *p* < 0.01.

**Figure 10 ijms-25-05621-f010:**
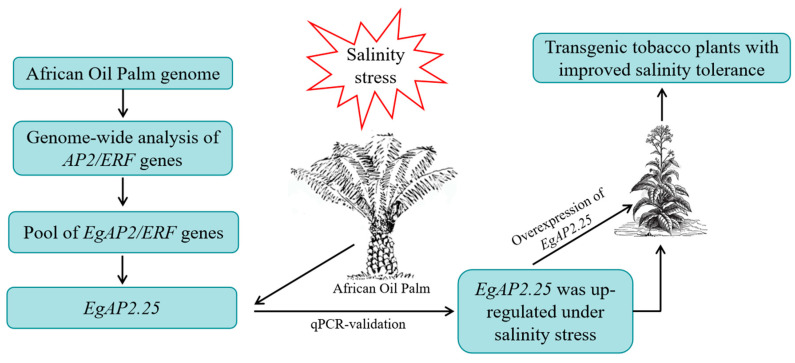
Schematic representation of the background information of the *EgAP2.25* gene.

## Data Availability

The data presented in this study are available on request from the corresponding author.

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
