# Peer review of "Oil Palm AP2 Subfamily Gene EgAP2.25 Improves Salt Stress Tolerance in Transgenic Tobacco Plants"

_ijms, 2024, doi:10.3390/ijms25115621_

Round 1

Reviewer 1 Report

Comments and Suggestions for Authors

Global climate change and its secondary consequences (such as soil salinity) have become the most important agricultural problem of today and the future. In this context, in the present article, the trend of salt tolerance by gene transfer in an agricultural crop with high economic value such as tobacco was examined in detail.

The main question that researchers are trying to address in the study is to understand the mechanism of salt stress tolerance of oil palm and to determine the role of AP2/ERF genes in other crops. The study also aimed to clone the EgAP2.25 gene from oil palm and analyze its sequence characterization and expression patterns.

Nowadays, many different methods are being tested to increase the tolerance of crops to abiotic stresses. Genetic transformation studies are sometimes used in this field. However, this study, which evaluated palm oil, AP2/ERF transcription factor family genes associated with this oil, and tobacco together, has closed a gap in the scientific world. GM tobacco plants transformed with this gene were reported to be less affected than control plants under salt stress at high concentrations, making this study valuable.

It is an original contribution to investigate the activity of this gene family in GM tobacco plants in terms of physiological and seed germination capacity.

The methodology used in the study is appropriate for the research objectives. However, the introduction is expected to give some essential information about the responses of tobacco plants to salt stress and the threshold value for damage.

In the study, the results are consistent with the evidence and arguments presented.  Cloning and sequence analysis of the EgAP2.25 gene using the palm plant was completed at the beginning of the study. Then, the effect of this gene family on salt tolerance was examined by various analyses starting from germination to antioxidant enzyme synthesis at later stages in GM tobacco plants. In the last stage, the usability of this gene as a marker in salt tolerance studies was monitored.

The references used in the study are appropriate in terms of both subject matter and relevance. In the study, data were shared using graphics and photographs, not tables. I think this is the most appropriate data-sharing system for this type of study.

 The most important deficiency in the whole study is the lack of a few sentences of information in the introduction section that evaluates the phenomena of tobacco and soil salinity together. 

Author Response

Dear editors and anonymous reviewers

We are glad to receive your valuable comments and suggestions to our manuscript. Thank you very much for your kind consideration on this manuscript “The oil palm AP2 subfamily gene EgAP2.25 improve salt stress tolerance in transgenic tobacco plants”. Without your professional reviews, this manuscript would not be as smooth as what it is now. Thank you very much! 

This is to confirm that we have amended the manuscript according to all the opinions, suggestions and comments and all the changes have been marked-up in the text by the red colored fonts. The responses to all the comments and suggestions are itemized as follows:

Reviewer 1

Comment 1:

Global climate change and its secondary consequences (such as soil salinity) have become the most important agricultural problem of today and the future. In this context, in the present article, the trend of salt tolerance by gene transfer in an agricultural crop with high economic value such as tobacco was examined in detail.

The main question that researchers are trying to address in the study is to understand the mechanism of salt stress tolerance of oil palm and to determine the role of AP2/ERF genes in other crops. The study also aimed to clone the EgAP2.25 gene from oil palm and analyze its sequence characterization and expression patterns.

Nowadays, many different methods are being tested to increase the tolerance of crops to abiotic stresses. Genetic transformation studies are sometimes used in this field. However, this study, which evaluated palm oil, AP2/ERF transcription factor family genes associated with this oil, and tobacco together, has closed a gap in the scientific world. GM tobacco plants transformed with this gene were reported to be less affected than control plants under salt stress at high concentrations, making this study valuable.

It is an original contribution to investigate the activity of this gene family in GM tobacco plants in terms of physiological and seed germination capacity.

The methodology used in the study is appropriate for the research objectives. However, the introduction is expected to give some essential information about the responses of tobacco plants to salt stress and the threshold value for damage.

In the study, the results are consistent with the evidence and arguments presented.  Cloning and sequence analysis of the EgAP2.25 gene using the palm plant was completed at the beginning of the study. Then, the effect of this gene family on salt tolerance was examined by various analyses starting from germination to antioxidant enzyme synthesis at later stages in GM tobacco plants. In the last stage, the usability of this gene as a marker in salt tolerance studies was monitored.

The references used in the study are appropriate in terms of both subject matter and relevance. In the study, data were shared using graphics and photographs, not tables. I think this is the most appropriate data-sharing system for this type of study.

 The most important deficiency in the whole study is the lack of a few sentences of information in the introduction section that evaluates the phenomena of tobacco and soil salinity together. 

Response: Thanks for your useful and professional comments. We have added information in the introduction section that evaluates the phenomena of tobacco and soil salinity together. 

We have amended the manuscript according to all your suggestions. Thanks again for your quick processing and professional editing of this manuscript. What you have done will always be highly and greatly appreciated. Any questions, we will be more than happy to answer. Looking forward to hearing from you soon. Best wishes!

  Coconut Research Institute, Chinese Academy of Tropical Agricultural Science, Hainan China

                                                  Rui Li and Jianqiu Ye  

                                                           2024-5-16

Reviewer 2 Report

Comments and Suggestions for Authors

There is no doubt that studies aimed at studying the functions of genes that determine plant resistance to biotic and abiotic stresses, and especially transcription factor genes, are of great importance. The work of respected authors on studying the function of EgAP2.25 gene in oil palm under salt stress response is undoubtedly relevant and will be of interest to a wide range of scientists.
The article is interestingly written and well illustrated. The illustrations are of high quality and complement the text well.  The conclusions in the article are consistent with the results obtained.
The research was done at a high level and I really liked the article.

I have only one suggestion for the authors - I think it would be better to add to the “introduction” a rationale for why exactly tobacco was chosen for transformation. Have there been other similar works? Can the new genetic background affect the functioning of the transferred genes, especially transcription factors, and affect the results of the experiment?

In general, I believe that the article can be accepted for publication after making minor corrections.

Author Response

Dear editors and anonymous reviewers

We are glad to receive your valuable comments and suggestions to our manuscript. Thank you very much for your kind consideration on this manuscript “The oil palm AP2 subfamily gene EgAP2.25 improve salt stress tolerance in transgenic tobacco plants”. Without your professional reviews, this manuscript would not be as smooth as what it is now. Thank you very much! 

This is to confirm that we have amended the manuscript according to all the opinions, suggestions and comments and all the changes have been marked-up in the text by the red colored fonts. The responses to all the comments and suggestions are itemized as follows:

Reviewer 2

Comment 1:

There is no doubt that studies aimed at studying the functions of genes that determine plant resistance to biotic and abiotic stresses, and especially transcription factor genes, are of great importance. The work of respected authors on studying the function of EgAP2.25 gene in oil palm under salt stress response is undoubtedly relevant and will be of interest to a wide range of scientists.

The article is interestingly written and well illustrated. The illustrations are of high quality and complement the text well.  The conclusions in the article are consistent with the results obtained.

The research was done at a high level and I really liked the article.

I have only one suggestion for the authors - I think it would be better to add to the “introduction” a rationale for why exactly tobacco was chosen for transformation. Have there been other similar works? Can the new genetic background affect the functioning of the transferred genes, especially transcription factors, and affect the results of the experiment?

In general, I believe that the article can be accepted for publication after making minor corrections.

Response: Thanks for your useful and professional comments. We have added information for choosing tobacco as a transgenic material in introduction part, and other researchers have also chosen tobacco for verification of AP2/ERF gene function.

We have amended the manuscript according to all your suggestions. Thanks again for your quick processing and professional editing of this manuscript. What you have done will always be highly and greatly appreciated. Any questions, we will be more than happy to answer. Looking forward to hearing from you soon. Best wishes!

  Coconut Research Institute, Chinese Academy of Tropical Agricultural Science, Hainan China

                                                   Rui Li and Jianqiu Ye  

                                                           2024-5-14

Reviewer 3 Report

Comments and Suggestions for Authors

The authors of the present manuscript have cloned the EgAP2.25 gene from E. guineensis Jacq. and have made its sequence characterization and expression analysis. Assessing numerous endpoints (antioxidant enzyme activities, osmolytes, chlorophyll content, etc.) in the transgenic tobacco plants affected by different concentrations of NaCl solution, the authors have obtained that over-expressing of EgAP2.25 could improve tolerance of plants to salinity stress.

The manuscript is interesting and well done. The English is easy to read and the work is understandable.

The title exactly reflects the main idea of the study and the content of the article.

The methods are fully appropriate and are accurately explained.

The results and discussion are well presented. The conclusions are well done.

 I have only small remarks.

  1. On line 48, page 2, the there is a mistype “(Figure 1a andc)”.
  2. The Figures 1, 6, 7 and 9 should be enlarged.
  3. In the legend of the Figure 3 has a grammar mistake.
  4. On lines 4-5, page 5, the following sentence should be corrected. “There was significant difference in dry/fresh weight between WT and transgenic plants under 0 mM NaCl (H2O).” In the present case, no significant difference was obtained.
  5. Please, revise the following sentence on lines 11-13, page 8: “AP2/ERF is a kind of superfamily transcription factor widely existing in plants, which plays an important regulatory role in regulating plant responses to abiotic stress and stimulates the interest for  researchers[19].”
  6. On lines 27 and 32, page 10, the name of the following bacterial species “Agrobacterium tumefaciens” should be given in italic.

Author Response

Dear editors and anonymous reviewers

We are glad to receive your valuable comments and suggestions to our manuscript. Thank you very much for your kind consideration on this manuscript “The oil palm AP2 subfamily gene EgAP2.25 improve salt stress tolerance in transgenic tobacco plants”. Without your professional reviews, this manuscript would not be as smooth as what it is now. Thank you very much! 

This is to confirm that we have amended the manuscript according to all the opinions, suggestions and comments and all the changes have been marked-up in the text by the red colored fonts. The responses to all the comments and suggestions are itemized as follows:

Reviewer 3

The authors of the present manuscript have cloned the EgAP2.25 gene from E. guineensis Jacq. and have made its sequence characterization and expression analysis. Assessing numerous endpoints (antioxidant enzyme activities, osmolytes, chlorophyll content, etc.) in the transgenic tobacco plants affected by different concentrations of NaCl solution, the authors have obtained that over-expressing of EgAP2.25 could improve tolerance of plants to salinity stress.

The manuscript is interesting and well done. The English is easy to read and the work is understandable.

The title exactly reflects the main idea of the study and the content of the article.

The methods are fully appropriate and are accurately explained.

The results and discussion are well presented. The conclusions are well done.

 I have only small remarks.

Comment 1:On line 48, page 2, the there is a mistype “(Figure 1a andc)”.

Response: Thanks for your kind words. We have revised it accordingly.

Comment 2:The Figures 1, 6, 7 and 9 should be enlarged.

Response: Thanks for your kind words. We have enlarged it accordingly.

Comment 3: In the legend of the Figure 3 has a grammar mistake.

Response: Thanks for your kind words. We have corrected the grammar mistake.

Comment 4:On lines 4-5, page 5, the following sentence should be corrected. “There was significant difference in dry/fresh weight between WT and transgenic plants under 0 mM NaCl (H2O).” In the present case, no significant difference was obtained.

Response: Thanks for your kind words. We have corrected the mistake. “There was no significant difference in dry/fresh weight between WT and transgenic plants under normal watering conditions”.

Comment 5:Please, revise the following sentence on lines 11-13, page 8: “AP2/ERF is a kind of superfamily transcription factor widely existing in plants, which plays an important regulatory role in regulating plant responses to abiotic stress and stimulates the interest for  researchers[19].”

Response: Thanks for your kind words. We have corrected the sentence. “The AP2/ERF superfamily is one of the largest gene families in plants, and it has at least one AP2 domain composed of around 60 amino acids [19].”

Comment 6:On lines 27 and 32, page 10, the name of the following bacterial species “Agrobacterium tumefaciens” should be given in italic.

Response: Thanks for your kind words. We have given it in italic. 

We have amended the manuscript according to all your suggestions. Thanks again for your quick processing and professional editing of this manuscript. What you have done will always be highly and greatly appreciated. Any questions, we will be more than happy to answer. Looking forward to hearing from you soon. Best wishes!

  Coconut Research Institute, Chinese Academy of Tropical Agricultural Science, Hainan China

                                                   Rui Li and Jianqiu Ye  

                                                           2024-5-14